# *Clostridioides difficile* Toxin B PCR Cycle Threshold as a Predictor of Toxin Testing in Stool Specimens from Hospitalized Adults

**DOI:** 10.3390/antibiotics11050576

**Published:** 2022-04-26

**Authors:** Sean Lee, Neha Nanda, Kenichiro Yamaguchi, Yelim Lee, Rosemary C. She

**Affiliations:** 1Department of Pathology, Keck School of Medicine of the University of Southern California, Los Angeles, CA 90033, USA; seanshlee41@gmail.com (S.L.); ky91789@gmail.com (K.Y.); 2Department of Medicine, Division of Infectious Diseases, Keck School of Medicine of the University of Southern California, Los Angeles, CA 90033, USA; neha.nanda@med.usc.edu; 3Department of Biology and Biological Sciences, California Institute of Technology, Pasadena, CA 91125, USA; ylee9912@gmail.com

**Keywords:** neutralization assay, toxin immunoassay, receiver operating characteristic curve

## Abstract

Rapid, accurate detection of *Clostridioides difficile* toxin may potentially be predicted by toxin B PCR cycle threshold (*tcdB* C_t_). We investigated the validity of this approach in an inpatient adult population. Patients who tested positive by *C. difficile* PCR (Cepheid GeneXpert) from December 2016 to October 2020 (*n* = 368) at a tertiary medical center were included. All stool samples were further tested by rapid glutamate dehydrogenase (GDH)/toxin B EIA and cell cytotoxin neutralization assay (CCNA). Receiver operating characteristic curves were analyzed. The area under the curve for *tcdB* C_t_ predicting toxin result by EIA was 0.795 (95% confidence interval (CI) 0.747–0.843) and by CCNA was 0.771 (95% CI 0.720–0.822). The Youden C_t_ cutoff for CCNA was ≤27.8 cycles (sensitivity 65.0%, specificity 77.2%). For specimens with C_t_ ≤ 25.0 cycles (*n* = 115), CCNA toxin was positive in >90%. The negative predictive value of *tcdB* C_t_ for CCNA was no greater than 80% regardless of cutoff chosen. In summary, very low C_t_ values (≤25.0) could have limited value as a rapid indicator of positive toxin status by CCNA in our patient population. A broad distribution of C_t_ values for toxin-negative and toxin-positive specimens precluded more robust prediction. Additional data are needed before broader application of C_t_ values from qualitatively designed assays to clinical laboratory reporting.

## 1. Introduction

*Clostridioides difficile* is an anaerobic, spore-forming Gram-positive bacillus and one of the most commonly reported pathogens in health care-associated infections [1]. In the context of a perturbed fecal microbiota, *C. difficile* causes disease via toxin production, leading to intestinal mucosal damage. Major risk factors for disease include prior antibiotic usage, older age, and healthcare exposure. The spectrum of disease ranges from diarrhea to pseudomembranous colitis and toxic megacolon. Both toxins A and B are produced by most pathogenic strains, but toxin B is detected in nearly all cases of *C. difficile* disease. Diagnosis is based upon the clinical suspicion and detection of toxigenic *C. difficile* or its toxins in stool [2].

The rapid, accurate diagnosis of *C. difficile* infection (CDI) is not yet fully optimized, but toxin detection may be considered the strongest correlate with clinical outcomes [3]. Methods to detect toxin B in stool include enzyme immunoassay (EIA), which has variable levels of performance [2,4], and cell culture cytotoxicity neutralization assay (CCNA). The detection of toxins correlates with disease severity [5], and CCNA results have been shown to correlate most closely with CDI compared to EIA-based toxin assays and toxigenic culture [3]. However, as CCNA is a time-consuming, with a manual method that requires up to 72 h for final results, surrogate methods have been put forth to hasten the time to an accurate toxicology result.

It has been demonstrated that the bacterial load of toxigenic *C. difficile* in stool correlates with the detection of toxins, with higher bacterial loads observed in specimens that test toxin-positive than those that test toxin-negative [6,7]. Several studies have therefore evaluated the cycle threshold (C_t_) from real-time PCR amplification of *C. difficile tcdB* from stool as a potentially rapid predictor of toxin status [6,7,8,9]. In our clinical experience, toxin EIA has performed poorly compared to CCNA [10], and we have not observed an obvious correlation between *tcdB* C_t_ and toxin status. It was therefore suspected that the predictive ability of *tcdB* C_t_ values may not be broadly applicable to different toxin assays or patient populations. The objective of this study was to investigate the potential use of *tcdB* C_t_ values in a hospitalized adult population for predicting toxin status by either toxin EIA or CCNA.

## 2. Results

### 2.1. Patient Demographics

Of 370 PCR-positive samples from hospitalized inpatients, 2 were excluded because CCNA was not performed due to lab error. The remaining 368 samples were from 191 (51.9%) male and 177 (48.1%) female patients (Appendix A). Mean and median ages were 58.7 and 62.0 years, respectively. Reasons for admission were largely related to patient history of solid organ transplant (*n* = 64 (17.4%)), hematopoietic stem cell transplant (*n* = 23 (6.3%)), malignancy (*n* = 105 (28.5%)), and surgical procedures (*n* = 97 (20.4%)). Underlying conditions of all patients are summarized in Table 1. 

### 2.2. Summary Statistics

Out of the 368 toxigenic *C. difficile* PCR-positive specimens, 326 (88.6%) tested positive by GDH EIA, 127 (34.5%) by toxin EIA, and 254 (69.0%) by CCNA. Compared to CCNA as the reference standard, toxin EIA had a sensitivity of 48.4% (123/254; 95% confidence interval (CI) 42.1–54.8%) and specificity of 96.5% (110/114; 95% CI 91.3–99.0%). The tcdB C_t_ values of the toxin EIA-positive, CCNA-negative specimens ranged from 26.1 to 35.0. Distribution of results demonstrated CCNA toxin-positive specimens to have a more gradual decline in numbers as C_t_ values increased, compared to toxin EIA-positive samples which demonstrated a denser clustering at lower C_t_ values (Figure 1). 

### 2.3. Cycle Threshold Value and GDH, Toxin EIA, and CCNA Results

The tcdB C_t_ values were significantly higher for GDH-negative than GDH-positive samples, toxin EIA-negative than toxin EIA-positive samples, CCNA-negative than CCNA-positive samples, and EIA-positive than CCNA-positive samples. There was no statistically significant difference between NAP1-negative and NAP1-presumptive positive samples (Table 2, Figure 2). However, NAP1-presumptive positive specimens were significantly more frequently EIA-positive (31/53; 58.5%) than NAP1-presumptive negative specimens (95/315; 30.2%) (*p* = 0.0001); and more frequently CCNA-positive (44/53; 83.0%) than NAP-1 presumptive negative specimens (210/315; 66.7%) (*p* = 0.016).

### 2.4. Use of tcdB C_t_ Value as an Indicator of Toxin Results

An ROC curve analysis of tcdB C_t_ values to predict toxin EIA results yielded an AUC of 0.795 (95% CI 0.747–0.843) (Figure 3). The Youden C_t_ cutoff of ≤26.2 cycles had a sensitivity of 75.6% (95% CI 67.4–82.2%) and specificity of 75.1% (95% CI 69.2–80.1%). The ROC curve for tcdB C_t_ values to predict CCNA toxin result yielded an AUC of 0.771 (95% CI 0.720–0.822) (Figure 3). The Youden C_t_ cutoff of ≤27.8 cycles had a sensitivity of 65.0% (95% CI 58.9–70.6%) and specificity of 77.2% (95% CI 68.7–83.9%). To account for the rapid turnaround time and accuracy of toxin EIA-positive results, we performed a subset analysis on toxin EIA-negative specimens, for which CCNA toxin results were potentially more applicable. In toxin EIA-negative specimens, the AUC was 0.677 (95% CI 0.610–0.745).

When examining the positive predictive value of tcdB C_t_ for CCNA toxin results by C_t_ value, we found that tcdB C_t_ ≤ 21.3 cycles was the highest cutoff at which positive toxin detection by CCNA could be predicted with 100% accuracy (*n* = 22). A cutoff of C_t_ ≤ 25.0 cycles was the highest at which >90.0% (104/115) of such specimens tested positive for toxin by CCNA. No meaningful ≥ C_t_ cutoff could predict negative CCNA toxin results beyond 80% accuracy. Even at a cutoff of ≥35.4 cycles, 4 of 19 (21.1%) specimens still tested CCNA-positive.

## 3. Discussion

Our analysis of data from a recent four-year period sought to characterize the predictive value of the tcdB C_t_ value for *C. difficile* toxin status in PCR-positive fecal specimens in an adult inpatient population. Our findings could be relevant to institutions considering the use of algorithmic or combination testing for *C. difficile* by toxin assays and PCR, particularly as we correlated PCR C_t_ values with CCNA, a reference standard toxin assay [11,12]. The rapid toxin EIA test used here can provide results within minutes, negating much of the benefit of using for C_t_ value from PCR to predict its results. Conversely, the low sensitivity for *C. difficile* toxin B by EIA compared to CCNA (48.4% in this study) limits its utility as a rapid toxin assay. Furthermore, use of the tcdB C_t_ value to predict toxin results obtained by CCNA could be considered more impactful given that CCNA toxin results have been shown to correspond with *C. difficile* disease severity [2,5], and that tcdB C_t_ values are obtained at the time of real-time PCR but CCNA requires 1 to 3 days.

In this study, *tcdB* C_t_ values yielded similar AUCs for toxin results by EIA and CCNA, and it was shown that using *tcdB* C_t_ value as predictor of toxin results yielded suboptimal sensitivity and specificity (~75%) at the optimal cutoffs. Selecting a separate cutoff *tcdB* C_t_ value for positive and negative toxin results by CCNA offered some advantages, albeit limited. Although a cutoff of C_t_ ≤ 25.0 cycles could predict positive CCNA results for a specimen with >90% accuracy, more than half of CCNA-positive specimens actually had Ct numbers >25.0. In our analysis, there no practical *tcdB* C_t_ cutoff value was found that reliably corresponded to CCNA-negative results given the wide and even distribution of CCNA-positive specimens across C_t_ values.

Of note, other studies evaluating the ability of tcdB C_t_ results to predict toxin status showed better performance than found here. An AUC as high as 0.921 for predicting combined results of toxin testing by EIA and CCNA using the Xpert assay, quantitatively calibrated to tcdB target concentrations, has been demonstrated (7). A sensitivity of 99.0% for rapid EIA toxin detection was attained with a tcdB C_t_ cutoff of <27.55 (Xpert), although the corresponding specificity was 58.8% [8]. It is noteworthy that in these two aforementioned studies, toxigenic bacterial load clustered tightly according to toxin test result, in contrast to our tcdB C_t_ results, which were much more broadly distributed. Similar to our study, another study evaluating Xpert PCR results from a 6-year period described significant overlap of C_t_ values between EIA toxin-positive and -negative specimens [13]. We can only speculate that differences in C_t_ value distribution seen between studies could have resulted from studies of longer study periods capturing more variation in test operators and assay lot-to-lot differences. Discrepancies in performance characteristics for the same *C. difficile* toxin and PCR assays have furthermore been observed to occur between different geographic sites and strain types [2,14,15], potentially contributing to our observed results. The low sensitivity of the rapid GDH/toxin combination at our institution is described in other studies, though it contrasts with the performance found by others [10,16,17,18,19], a trend which remained consistent throughout this four-year study period. Our results were similar to those described in a cancer center patient population, in which an AUC of 0.83 was obtained with a Youden cutoff of ≤28.0 cycles (vs. our Youden cutoff of ≤27.8 cycles) for the prediction of CCNA toxin results. Overlapping distributions of Ct values of 25.0–28.0 were also noted between CCNA-negative and CCNA-positive cases in another comparison [20]. Although others have not found adverse outcomes associated with the implementation of reporting tcdB C_t_ values [21], it is important to note that test performance characteristics may differ by institution.

These studies demonstrate the potential applications of C_t_ values from the typically qualitative *C. difficile* PCR. The Xpert assay is generally considered to have excellent sensitivity in organism detection and specificity in ruling out CDI, similar to that of other commercial *C. difficle* PCR assays [22,23]. It is stressed that such application of C_t_ values is not FDA-cleared for *C. difficile* PCR testing, including the commonly used Xpert assay. Test cartridges are inoculated with a swab dipped into the fecal specimen; hence, the starting material is a non-standard, non-quantified amount. Quantitative results could further be subject to variations in operator technique, stool consistency, and DNA recovery, among other factors. To the contrary, one study found that variations in stool input volume had little bearing on the C_t_ value, and moreover, the coefficient of variation for the tcdB C_t_ was only 2.8% across four lots of the Xpert assay [8]. In additional support of the potential quantitative use of *C. difficile* PCR results, C_t_ values were found to significantly correlate with quantitative culture results for *C. difficile* in stool [6]. In our retrospective analysis of clinical test results, we did not calibrate C_t_ against a standard curve to mitigate any lot-to-lot variation over the study period. In order for C_t_ values for qualitative tests to offer maximal performance as a toxin prediction tool for clinical purposes, we believe the test should be managed in the laboratory as a quantitative, laboratory-developed test. Our study results otherwise advise against use of the C_t_ result at face value for clinical purposes.

In addition to predicting toxin status, the bacterial load of toxigenic *C. difficile* in stool has been proposed as a predictor of CDI severity and clinical outcome [9,23,24]. However, the actual value of its prognostic contribution, as compared to the information derived from toxin testing and assessment of clinical risk factors, has been questioned [24]. With different case definitions and specimen inclusion criteria between different studies, optimal cutoffs for tcdB C_t_ values from the Xpert platform have ranged from <23.5 to <27.55 for predicting poor outcomes for CDI [9,19,24,25,26,27]. Incorporating expert clinical consensus, a more recent study found a threshold of <24.0 cycles on the Xpert assay corresponded with a high probability of CDI [28].

Gene targets of the Xpert *C. difficile* PCR assay include binary toxin and tcdC deletion for presumptive detection of the NAP1 strain. Outside the epidemic setting, the clinical significance of the NAP1 strain is variable, with some studies noting increased associated CDI severity and mortality but others finding host factors to be more contributory to outcome than strain type [29,30,31]. In either case, NAP1 strains are known to produce higher levels of toxin A and B in vitro and in vivo [31,32]. Similar to others [7,24], this study observed a trend (*p* = 0.056) towards lower C_t_ values for presumptive NAP1 strains and higher frequency of toxin positivity by either EIA or CCNA compared to NAP1-negative specimens. These results were from a single center among adult inpatients, and the results may not be generalizable to other patient populations. CCNA testing was performed by a reference laboratory within specimen stability limits, rather than immediately set up for in-house testing, which may have had a minor impact on CCNA results. Another consideration is that only Bristol scale 6 or 7 stools were included for clinical testing to improve appropriate test utilization, although this restriction may have excluded some cases of CDI [33,34]. Because our purpose was to analyze the practical implications of using C_t_ as a simple indicator of toxin status, wherein the clinical laboratory would operate within its current test algorithm, we did not further stratify patients using clinical severity scores or other clinical variables to improve the predictive power of the tcdB C_t_ value for CCNA toxin status. This remains an important consideration to be further explored in the future.

In summary, we found that the use of tcdB C_t_ values from a commonly used *C. difficile* PCR platform was limited in its ability to predict CCNA toxin status. The difficulty was related to the broad distribution of tcdB C_t_ values observed in the study period, even among CCNA-positive specimens. Our data were compiled from several years of clinical testing and found that results were not as robust and reliable compared to focused studies occurring over a short period of time or those using calibrated assays.

## 4. Materials and Methods

### 4.1. Specimen Inclusion Criteria

In this retrospective analysis, we included the clinical results of all stool samples from hospitalized adult patients that had undergone *C. difficile* testing by EIA and CCNA (detailed below) after testing positive for *C. difficile* toxin gene *tcdB* by real-time PCR (GeneXpert *C. difficile*/Epi PCR, Cepheid, Sunnyvale, CA, USA), which, at the time of study, was from December 2016 to October 2020. The patients were from a tertiary care center specializing in surgical, transplant, and oncology care (Keck Medical Center, Los Angeles, CA, USA). Per institutional policy, specimens were rejected if not meeting Bristol stool scale type 6 or 7 or if a prior specimen had been tested from the same patient within the past 7-day period.

### 4.2. Data Collection and C. difficile Test Methods

*C. difficile* PCR testing was performed upon physician’s request based on clinical suspicion. *C. difficile* testing is performed promptly upon receipt by the laboratory. For hospitalized patients, standard-of-care testing for specimens positive by *C. difficile* PCR included immediate testing by a rapid EIA assay for glutamate dehydrogenase (GDH) and toxin B (Cdiff Quick Check, Alere Inc., Waltham, MA, USA), and immediately freezing an aliquot and sending it for CCNA testing at a reference laboratory (ARUP Laboratories, Salt Lake City, UT, USA). *C. difficile* test results by PCR, EIA, and CCNA, as well as *tcdB* C_t_ values and NAP1 results from the Xpert PCR assay, were retrospectively collected for this study. Electronic medical records were reviewed for patient demographics and underlying health conditions.

### 4.3. Statistical Analysis

Mean ranks of *tcdB* C_t_ values between groups were compared using the Mann–Whitney U test. Frequency of categorical variables between groups were compared using two-tailed Fisher’s exact test. Receiver operating characteristic (ROC) curve analysis was performed using C_t_ values as a marker of toxin positivity by either EIA or CCNA. The area under the curve (AUC) for each toxin test was calculated and compared using the Wilcoxon trapezoidal method. The optimal C_t_ cutoff value was determined using the Youden maximum index value. Statistical calculations were conducted using GraphPad Prism 9.0.0 (San Diego, CA, USA).

## 5. Conclusions

The application of tcdB C_t_ values from *C. difficile* PCR in a tertiary care, adult inpatient population was suboptimal as a predictive tool to determine *C. difficile* toxin status by the reference cytotoxin neutralization assay or by enzyme immunoassay. Substantial overlap of tcdB C_t_ values between toxin-negative and toxin-positive specimens precluded more accurate prediction.

## Figures and Tables

**Figure 1 antibiotics-11-00576-f001:**
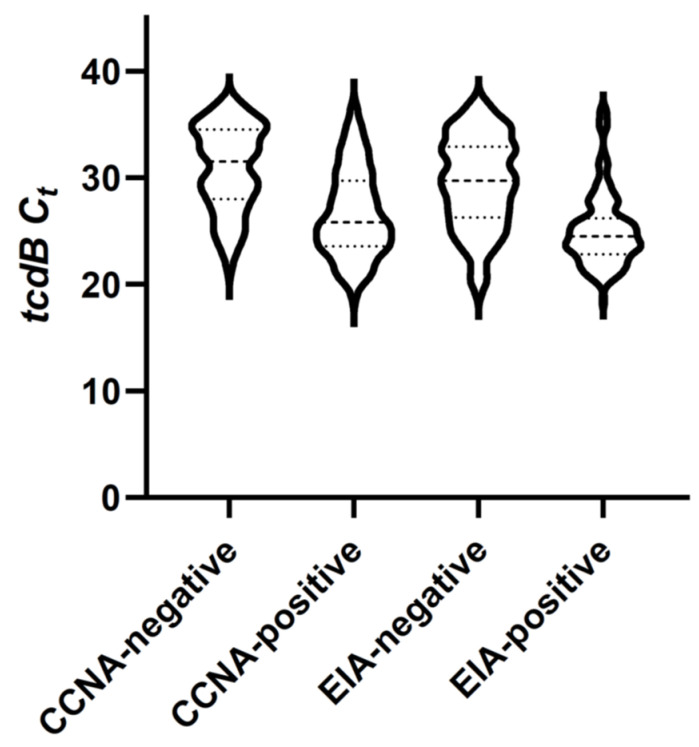
Box and violin plot shows distributions of *tcdB* C_t_ values according to toxin test results. Minimum, maximum, median (large dashed line), and 25th and 75th percentiles (small dashed lines) are indicated. Observed frequencies of values are represented by width of the plot interval.

**Figure 2 antibiotics-11-00576-f002:**
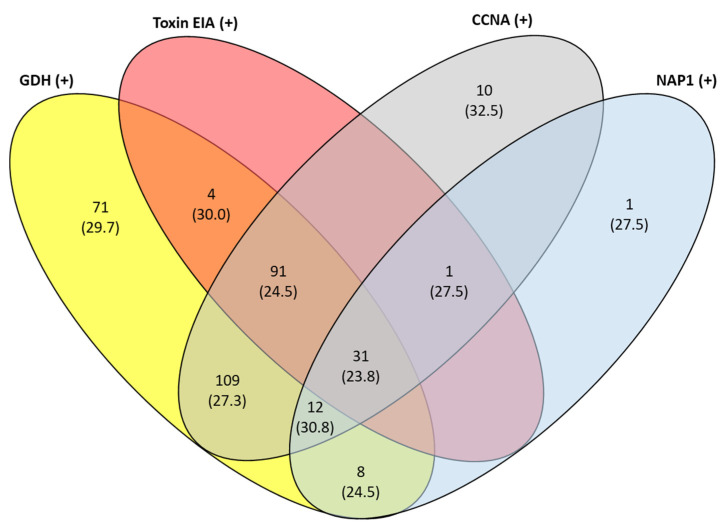
Venn diagram illustrates the number of specimens with each different test result combination, given as *n* (median tcdB C_t_ value). Abbreviations: C_t_, threshold cycle; GDH, glutamate dehydrogenase; EIA, enzyme immunoassay; CCNA, cell culture cytotoxicity neutralization assay; NAP1, North American PFGE type 1.

**Figure 3 antibiotics-11-00576-f003:**
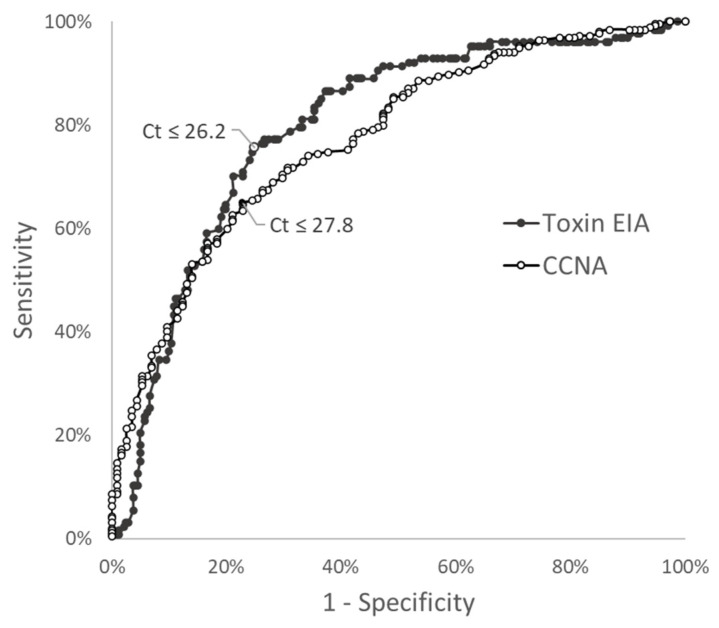
Receiver operating characteristic (ROC) curve for *tcdB* C_t_ value predicting EIA toxin status (AUC = 0.795), or CCNA toxin status (AUC = 0.771). Youden cutoffs are indicated on each curve with their corresponding C_t_ value cutoffs.

**Table 1 antibiotics-11-00576-t001:** Underlying medical conditions of hospitalized adult patients with positive *C. difficile* PCR included in this study.

Medical Condition	*n* (%)
Malignancy	105 (28.5)
Hematologic	15 (4.1)
Non-hematologic	90 (24.5)
Solid organ transplant	64 (17.4)
Hematopoietic stem cell transplant	23 (6.3)
Surgical procedure	97 (20.4)
Neurosurgery	22 (6.0)
Abdominal	29 (7.9)
Cardiovascular	26 (7.1)
Orthopedic	8 (1.6)
Urologic	4 (1.1)
Other	8 (2.2)
Cardiovascular disease	20 (5.4)
Hepatic failure	17 (4.6)
Inflammatory bowel disease	15 (4.1)
Gastrointestinal disease (non-surgical)	9 (2.4)
Non-cancerous neoplasm	5 (1.4)
Other conditions	13 (3.5)
Total	368 (100)

**Table 2 antibiotics-11-00576-t002:** Summary statistics of C_t_ values for *tcdB* by GDH EIA, toxin EIA, CCNA, and PCR NAP1 results.

	*n*	Median (Mean) C_t_	*p*-Value ^a^
GDH-positive	326	27.5 (27.3)	<0.001
GDH-negative	42	32.9 (32.7)	
Toxin EIA-positive	127	25.0 (24.5) ^b^	<0.001
Toxin EIA-negative	241	29.5 (29.7)	
CCNA-positive	254	25.8 (26.6) ^b^	<0.001
CCNA-negative	114	31.5 (30.9)	
NAP1-presumptive positive	53	25.3 (26.9)	0.056
NAP1-negative	315	27.7 (28.1)	
All samples	368	27.5 (27.9)	NA

^a^ Mann–Whitney U test. ^b^
*p* = 0.002 for comparison by Mann–Whitney U test. Abbreviations: C_t_, threshold cycle; GDH, glutamate dehydrogenase; EIA, enzyme immunoassay; CCNA, cell culture cytotoxicity neutralization assay; NAP1, North American PFGE type 1; NA, not applicable.

## Data Availability

The data presented in this study are available in the Appendix A.

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
