# Peer review of "Clostridioides difficile Toxin B PCR Cycle Threshold as a Predictor of Toxin Testing in Stool Specimens from Hospitalized Adults"

_antibiotics, 2022, doi:10.3390/antibiotics11050576_

Round 1

Reviewer 1 Report

This manuscript by Sean Lee et al. describes interesting comparative data on the methodology for diagnosing C. difficile infection (with toxins).
Suffering from a lack of novelty, the manuscript deserves to be modified before it is eventually accepted for publication.
Global: prefer passive voice.
Results: 
The main problem is related to the thermolability of the toxins for EIA. Thus, and in the absence of data on analysis time, the discordant results obtained with GDH alone should not be considered a limitation.
It is surprising that NACC is considered a gold standard, given that most laboratories have discontinued its use and that its performance is significantly inferior to that of PCR.
Table 2 could be effectively completed with a Venn diagram.
Italicize "vs.". and bacteria names.
The point made in 2.4 should be clearly indicated on the ROC. Also, the ROC figure could be summarized in one figure with all the tests, to better understand their difference.
The authors should discuss the difference in performance between GenXpert and other PCRs for C. difficile. In addition, they should discuss the potential impact of O27 and binary toxin on their data. 

Methods: 
How the number of stools to be included was determined.
The Bristol stool scale of 6 and 7 is quite high and the authors overlook potential diagnoses with a lower Bristol score. Discuss this and test them.

Author Response

Reviewer 1

Point 1: Global: prefer passive voice.

Response 1: The majority of the manuscript is written in the passive voice. We have converted some of the active voice sentences to the passive voice. If active voice is inappropriately used in particular areas, please specify the areas and we would be happy to revise it.

Point 2: Results:

The main problem is related to the thermolability of the toxins for EIA. Thus, and in the absence of data on analysis time, the discordant results obtained with GDH alone should not be considered a limitation.

Response 2:

Thank you for raising this point. We agree that thermolability is a pre-analytical issue with C. difficile toxin testing. We have added clarity to the Methods that specimens were tested promptly after receipt by the laboratory. Because this study includes only hospitalized patients, the transit time of stool specimens from bedside to the hospital laboratory is minimized and toxin lability is not an issue beyond that normally encountered in hospital laboratory settings. Our data demonstrated wide discrepancy between EIA and CCNA for toxin detection, thus toxin lability itself cannot explain its lack of toxin detection compared to GDH or PCR. We have also added references to demonstrate the sensitivity of the toxin EIA which we observed is within the range described by other studies.

Point 3: It is surprising that NACC is considered a gold standard, given that most laboratories have discontinued its use and that its performance is significantly inferior to that of PCR.

Response 3: We agree that the cumbersome nature of CCNA has led to its discontinuation by most clinical laboratories. Yet its status remains as a reference standard against which other assays are compared. We have added another reference to support the statement (Bagdasarian N, Rao K, Malani PN. Diagnosis and treatment of Clostridium difficile in adults: a systematic review. JAMA. 2015 Jan 27;313(4):398-408). Robust studies have demonstrated that of all diagnostic tests, CCNA correlates most highly with disease severity and prognosis, as cited in the manuscript. Providers are clinically concerned that PCR detection of organism may indicate colonization rather than true disease, therefore they also rely on toxin testing results by CCNA to aid in clinical decision making.

Point 4: Table 2 could be effectively completed with a Venn diagram.

Response 4: We have added a Venn diagram, Figure 2. Table 2 is kept to more effectively convey statistical comparisons.

Point 5: Italicize "vs.". and bacteria names.

Response 5: Thank you for noticing the formatting error. We have corrected the formatting of the bacterial names and genes.

Point 6: The point made in 2.4 should be clearly indicated on the ROC. Also, the ROC figure could be summarized in one figure with all the tests, to better understand their difference.

Response 6: It is somewhat unclear which point made in 2.4 should be clearly indicated on the ROC. Nevertheless, we have consolidated the 2 ROC curves into a single figure and added Youden cutoff points and their corresponding Ct value thresholds to the curves. We hope this adequately addresses the point.

Point 7: The authors should discuss the difference in performance between GenXpert and other PCRs for C. difficile. In addition, they should discuss the potential impact of O27 and binary toxin on their data.

Response 7:

We thank the reviewer for these suggestions. A statement has been added to the Discussion on the general performance of the GenXpert PCR as a well established clinical assay and its inclusion of NAP1 strain targets. In addition, the impact of NAP1 on toxin testing results has been analyzed and added to the Methods, Results, and Discussion.

Point 8: Methods:

How the number of stools to be included was determined.

Response 8: All data points from our population were included, e.g. from time of implementation of the C. difficile test methods at our institution (PCR, GDH, toxin EIA, and CCNA) up until the time the study was conducted were included. Data from the 4-year period was determined to be adequate for the purpose of exploring if the PCR cycle threshold value could provide clinical benefit. As this was a retrospective analysis of existing data, resources were not limiting therefore all 4 years were included. Method of determining number of stools is now added.

Point 9: The Bristol stool scale of 6 and 7 is quite high and the authors overlook potential diagnoses with a lower Bristol score. Discuss this and test them.

Response 9: We have now added to the Discussion on the restriction of testing to Bristol stool 6 and 7 specimens, for which there are advantages and disadvantages. We also include two references supporting the issues including those raised by the Reviewer. Our hospital policy (as is common in the U.S.) has been to restrict C. difficile testing to Bristol scale 6 and 7 specimens in an effort to reduce overdiagnosis, improve antimicrobial stewardship, and improve test utilization. It is therefore unfortunately not possible in this retrospective analysis to test more formed specimens.

Reviewer 2 Report

The author summarized  investigate the potential use of tcdB Ct values in our hospitalized adult population for predicting toxin status by either toxin EIA or CCNA.  Authors listed  ability of tcdB Ct values may not be broadly applicable to different toxin assays or patient populations.

The manuscript is well-written. Overall, the manuscript has provided good structure by good order of subheadings.

  • Introduction is very short.
  • Table 1: Remove total after %.
  • Add conclusion at the end of article.
  • References need to increase and updated

Author Response

Reviewer 2

Point 1: Introduction is very short.

Response 1: We have expanded the Introduction, as also requested by Reviewer 3.

Point 2: Table 1: Remove total after %.

Response 2: “Total” is now removed.

Point 3: Add conclusion at the end of article.

Response 3: Thank you for the suggestion. A conclusion has been added.

Point 4: References need to increase and updated

Response 4: We agree with this. We have updated and increased our number of references to 35.

Reviewer 3 Report

The work presents some interesting results in a clear and direct way.
The manuscript contains a series of small errors that do not detract from the merit of the work but do diminish its quality and presentation and it is necessary to correct them before being accepted.
The introduction is direct and very short. A little more should be said about the Clostridioides difficile bacterium, saying that it is a great positive bacterium and explaining well what are the causes that lead a hospitalized person to be infected with this bacterium and indicate the damage that it can produce in the colon and pass to explain its toxins A and B. The title has toxin A included and it is not mentioned at all in the introduction. The introduction is focused on the technical part of the proposed methodology, but other aspects are left that can make the work more attractive for the readers of the journal and for the work to be cited. The introduction must clearly contain the objective of the work. The main objective was…
Line 63: toxigenic C. difficile PCR… Line 110: for C. difficile toxin status, line 112: C. difficile by toxin, line 116: for C. difficile toxin B. The scientific name of the bacteria must always be in italics correct throughout the manuscript.
Lanes 91-96: ROC curve analysis of tcdB Ct values ​​to predict toxin EIA results yielded an AUC of 91 0.795 (95% CI 0.747-0.843) (Figure 2A). The Youden Ct cutoff of ≤26.2 cycles had a sensitivity of 75.6% (95% CI 67.4-82.2%) and specificity of 75.1% (95% CI 69.2-80.1%). The 93 ROC curve for tcdB Ct values ​​to predict CCNA toxin resulted yielded an AUC of 0.771 94 (95% CI 0.720-0.822) (Figure 2B). The Youden Ct cutoff of ≤27.8 cycles had a sensitivity of 95 65.0% (95% CI 58.9-70.6%) and specificity of 77.2% (95% CI 68.7-83.9%). A criterion must be unified to express numerical data. Some figures have one decimal place, others two, others three. Correct the entire manuscript.
Review the format of bibliographic references so that it is unified.

Best regards,

Author Response

Reviewer 3

Point 1: The introduction is direct and very short. A little more should be said about the Clostridioides difficile bacterium, saying that it is a great positive bacterium and explaining well what are the causes that lead a hospitalized person to be infected with this bacterium and indicate the damage that it can produce in the colon and pass to explain its toxins A and B. The title has toxin A included and it is not mentioned at all in the introduction. The introduction is focused on the technical part of the proposed methodology, but other aspects are left that can make the work more attractive for the readers of the journal and for the work to be cited. The introduction must clearly contain the objective of the work. The main objective was…

Response 1:

We thank Reviewer 3 for the careful review of the manuscript. The introduction has been expanded to include all of the points made by the Reviewer, including organism characteristics, risk factors for disease, toxin mediation of disease, and objective of the work. To clarify, we do not see that Toxin A is included in the Title and presume the Reviewer meant Toxin B.

Point 2: Line 63: toxigenic C. difficile PCR… Line 110: for C. difficile toxin status, line 112: C. difficile by toxin, line 116: for C. difficile toxin B. The scientific name of the bacteria must always be in italics correct throughout the manuscript.

Response 2:

We apologize for the formatting oversight. Bacterial names have now all been italicized.

Point 3: Lanes 91-96: ROC curve analysis of tcdB Ct values ​​to predict toxin EIA results yielded an AUC of 0.795 (95% CI 0.747-0.843) (Figure 2A). The Youden Ct cutoff of ≤26.2 cycles had a sensitivity of 75.6% (95% CI 67.4-82.2%) and specificity of 75.1% (95% CI 69.2-80.1%). The ROC curve for tcdB Ct values ​​to predict CCNA toxin resulted yielded an AUC of 0.771 94 (95% CI 0.720-0.822) (Figure 2B). The Youden Ct cutoff of ≤27.8 cycles had a sensitivity of 65.0% (95% CI 58.9-70.6%) and specificity of 77.2% (95% CI 68.7-83.9%). A criterion must be unified to express numerical data. Some figures have one decimal place, others two, others three. Correct the entire manuscript.

Response 3: Respectfully, the number of significant digits for all of the data provided is already standardized as 3. The Ct values are those provided by the instrument and should also be considered separately from other calculations in terms of decimal places.  

Point 4: Review the format of bibliographic references so that it is unified.

Response 4: The format of the References has been corrected to standard across all citations.

Round 2

Reviewer 1 Report

The manuscript has been improved, thanks to the work of the authors.